# Promoting Circular Economy in the Palm Oil Industry through Biogas Codigestion of Palm Oil Mill Effluent and Empty Fruit Bunch Pressed Wastewater

Chaisri Suksaroj [1], Kanokwan Jearat [2], Nutthayus Cherypiew [3], Cheerawit Rattanapan [4] and Thunwadee Tachapattaworakul Suksaroj [4,*]

[1]  Research Center for Sustainable Development, Department of Irrigation Engineering, Faculty of Engineering at Kamphaeng Saen, Kasetsart University, Kamphaeng Saen Campus, Nakhon Pathom 73140, Thailand; chaisri.s@ku.th
[2]  Faculty of Management Sciences, Nakhon Si Thammarat Rajabhat University, Tha Ngio, Muang Nakhon Si Thammarat 80280, Thailand; kanokwan.micro19@gmail.com
[3]  Environmental Management Program, Faculty of Environmental Management, Prince of Songkla University, Songkla 90112, Thailand; nutthatus_pto@hotmail.com
[4]  ASEAN Institute for Health Development, Mahidol University, Salaya, Putthamonthon, Nakhon Pathom 73170, Thailand; cheerawit.rat@mahidol.edu
*  Correspondence: thunwadee.suk@mahidol.edu

**Abstract:** This research aimed to investigate the biogas production and circular economy perspective in the palm oil industry through codigestion of oil palm empty fruit bunch (EFB) pressing wastewater and palm oil mill effluent (POME). The EFB pressing method constitutes an alternative new technology used to extract the remaining oil, increasing palm oil product; however, it produces highly polluted wastewater. Batch experiments were carried out at 35 °C to investigate the optimal ratios of EFB wastewater, inoculums, and POME. The optimal condition was 45% POME + 50% seed + 5% EFB wastewater. This condition was then used in semicontinuous fermentation where the optimal hydraulic retention time (HRT) totaled 25 days. The accumulated biogas was 18,679 mL/L while the accumulated methane totaled 6778 mL/L. The methane content was 62%, and the COD removal efficiency was 67%. The sludge produced from the HRT 25-days digester complied with the organic compost standard which could be further used to nourish the soil. An economic analysis of the EFB pressing project revealed a higher internal rate ratio with shorter payback compared with the conventional process. These results provide information on the circular economic approach to promote sustainable palm oil processing.

**Keywords:** biogas; palm oil industry; circular economy; clean energy; sustainability

## 1. Introduction

The ASEAN (Association of Southeast Asian Nations) Region stands as the primary hub for palm oil production, given its position as the largest producer. The palm oil industry holds significant importance, especially in the Republic of Indonesia and Malaysia. These two countries have emerged as the top global primary producers and exporters, accounting for 56 and 30% of the world's palm oil supply, respectively. Additionally, Thailand secures its place as the fourth largest palm oil exporter globally. Although Thailand's contribution to global palm oil supply is approximately 2% [1], it remains a crucial maincrop that significantly contributes to the economic growth of both the country and the entire region. The typical process for extracting palm oil is a wet method using a large amount of water in the production process, resulting in generating palm oil effluent (POME) at a rate of about 0.7 to 0.9 cubic meters per ton of fresh palm fruit [2,3]. POME is a nonharmful waste; however, it will pose environmental issues because of the vast oxygen draining capacity in oceanic frameworks because of natural and supplemental substances. The wastes are in

the form of high organic matter concentration, such as cellulosic wastes with a mixture of carbohydrates and oils. This effluent has a dark brown color and contains organic matter at a concentration of over 20,000 mg/L [4]. The discharge of untreated POME creates adverse impacts to the environment. Currently, anaerobic action is commonly used to treat POME, efficiently removing organic matter and producing biogas as a renewable energy source [5,6].

Additionally, aside from POME, palm oil processing generates various waste products, including oil palm trunks (OPT), oil palm fronds (OPF), empty fruit bunches (EFBs), palm pressed fibers (PPF), palm kernel shells, and less fibrous material, such as palm kernel cake [7]. Among these waste materials, a substantial amount and mass are attributed to fresh EFBs. For every 100 kg of oil palm fresh fruit bunches processed for oil production, approximately 30 to 60 kg of oil palm EFBs are discarded as waste. The improper disposal of EFBs on land leads to pollution in the surrounding areas, as these EFBs still contain oil that can contaminate the local environment. To address this issue, composting has emerged as a favored alternative for managing solid waste in Thailand and other countries. Due to the residual oil content in the EFB, palm oil factories in Thailand now employ compression techniques to extract the remaining oil, resulting in increased productivity and providing a higher yield of compostable fiber after the re-pressing process.

Wastewater obtained from EFB pressing exhibits higher levels of chemical oxygen demand (COD), biological oxygen demand (BOD), suspended solids (SS), and various substances compared with the typical wastewater produced during crude palm oil extraction processes. These characteristics make it a viable resource for biogas production and energy generation. However, if the EFB pressing wastewater is introduced directly in the fermentation process, it could pose significant toxicity issues for microorganisms, potentially leading to system failure. Thus, it becomes crucial to mitigate its toxicity by diluting it with general wastewater from the palm oil extraction plant. Furthermore, conducting a cost analysis of this alternative process is essential to determine its feasibility and encourage wider application. Hence, this research aimed to investigate the production of biogas from EFB pressing wastewater through codigestion with general wastewater from the palm oil extraction plant. This study focused on analyzing the composition of EFB pressing wastewater and evaluated the effectiveness of its codigestion with general wastewater in enhancing biogas production and methane content. The findings of this study can serve as a foundation for informed decision making regarding integrating EFB pressing in palm oil extraction and this wastewater feeding in biogas production. This approach has the potential to optimize wastewater management, minimize resource consumption in renewable energy production, and enhance the value of waste generated during the palm oil extraction process.

## 2. Materials and Methods

### 2.1. Substrates

The characteristics of POME and EFB pressed wastewater were collected from the receiving tanks of a palm oil factory using the grab sampling method. The inoculum used for the study was collected from the anaerobic wastewater treatment plant of the same factory and stored at a temperature of 4 °C until analysis. The collected samples were analyzed for their characteristics, including COD, BOD, pH, SS, total solids (TS), volatile solids (VS), alkalinity, volatile fatty acids (VFA), ammonia nitrogen ($NH_3$-N), total Kjeldahl nitrogen (TKN), mixed liquor SS (MLSS), and grease and oil (G&O), using the Standard Methods for the Examination of Water and Wastewater, 23rd Edition [8]. The characteristics of the pressing machine, EFB wastewater, and EFB before and after pressing are illustrated in Figure 1.

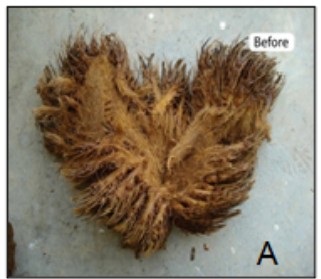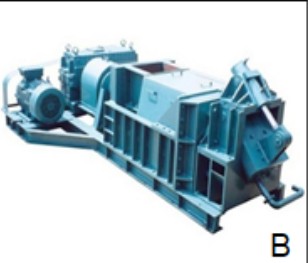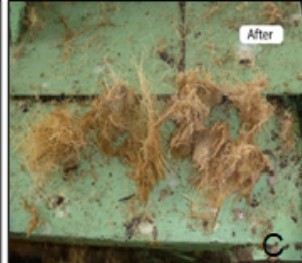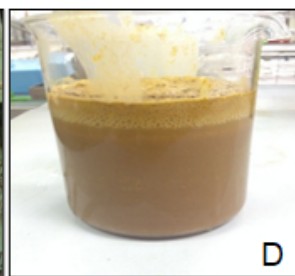

**Figure 1.** EFB before pressing (**A**), pressing machine (**B**), EFB after pressing (**C**), and EFB wastewater (**D**).

### 2.2. Laboratory-Scale Batch Reactor

The laboratory-scale reactor system consists of a glass bottle with a total volume of 1 L and a working volume of 0.5 L. The bottle was sealed with a rubber septum and covered with parafilm to ensure airtightness. The anaerobic fermentation process was conducted at a temperature of 35 °C with daily agitation by shaking the bottle once a day. The biogas produced in the bottle was displaced in a glass bottle filled with water to measure its volume daily. The gas composition was analyzed using gas chromatography (GC, Hewlett-Packard Model 6890, Palo Alto, CA, USA) every seven days. The percentage of methane analyzed was multiplied by the measured volume of total biogas to calculate the daily production of methane. The types and amounts of substrates used in the experiment are presented in Table 1.

**Table 1.** Types of substrates and the quantity of POME, seed inoculum, and EFB wastewater used in the batch-type biogas production experiment, totaling 500 mL.

| Set No. | Type of Substrate | POME (mL) | Inoculum (mL) | EFB Press (mL) | C:N |
|---|---|---|---|---|---|
| 1 | 65% POME + 35% seed | 325.00 | 175.00 | - | 140:1 |
| 2 | 62.5% POME + 35% seed + 2.5% EFB ww | 316.80 | 175.00 | 8.20 | 142:1 |
| 3 | 60% POME + 35% seed + 5% EFB ww | 308.75 | 175.00 | 16.25 | 135:1 |
| 4 | 55% POME + 35% seed + 10% EFB ww | 292.50 | 175.00 | 32.50 | 208:1 |
| 5 | 50% POME + 50% seed | 250.00 | 250.00 | - | 63:1 |
| 6 | 47.5% POME + 50% seed + 2.5% EFB ww | 243.75 | 250.00 | 6.25 | 116:1 |
| 7 | 45% POME + 50% seed + 5% EFB ww | 237.50 | 250.00 | 12.50 | 126:1 |
| 8 | 40% POME + 50% seed + 10% EFB ww | 225.00 | 250.00 | 25.00 | 147:1 |
| 9 | 25% POME + 75% seed | 125.00 | 375.00 | - | 74:1 |
| 10 | 22.5% POME + 75% seed + 2.5% EFB ww | 121.88 | 375.00 | 3.12 | 93:1 |
| 11 | 20% POME + 75% seed + 5% EFB ww | 118.75 | 375.00 | 6.25 | 117:1 |
| 12 | 15% POME + 75% seed + 10% EFB ww | 112.50 | 375.00 | 12.5 | 78:1 |

Note: ww—wastewater.

### 2.3. Semicontinuous Laboratory-Scale Reactor

The experimental setup for studying the production of biogas in a semicontinuous laboratory-scale reactor system consisted of a 4 L brown-colored glass bottle with a cylindrical shape as a fermenter, with a working volume of 3 L, and a 1 L gas storage system connected by a balloon tube to store the gas. The mixture was stirred using a small water pump to ensure adequate mixing inside the fermenter. The experiment was conducted in a semicontinuous completely stirred tank reactor (CSTR) system, controlling appropriate experimental conditions for anaerobic fermentation without air circulation, at a constant temperature of 35 ± 1 °C using a heater and a water bath to maintain the temperature level constant.

### 2.4. Codigestion Experiments

The study was conducted by codigesting normal wastewater from palm oil extraction and EFB wastewater to find the optimal ratio producing the highest methane yield. The

experiment was conducted in a batch laboratory system, and the results were used to guide the following experiment, which was conducted using a semicontinuous system.

### 2.4.1. Batch Experiments

The batch experiments were conducted using the following steps: all three types of fermentable materials were brought to room temperature before conducting the experiment, the three types of fermentable materials were separated and placed in a container, the volume of each fermentable material was measured and combined according to the ratio calculation of the COD/N (chemical oxygen demand/nitrogen) ratio [9], using cofermentable materials in the ratios of 2.5, 5, and 10% $v/v$, and the fermentation material was added to the anaerobic digestion system as a one-time feeding. Then, the starting inoculum was used at 20, 35, 50, and 75% of the working volume of the fermentation material to determine the appropriate proportion of the microorganisms in gas production (Table 1) as recommended by [10]. They suggested that the amount of inoculum should be no less than 10% of the working volume, and then the pH was adjusted in the range of 7.0 to 7.2. The anaerobic fermentation system was created from a 1 L glass bottle with a working volume of 0.5 L. The control set, including main materials mixed with different ratios of inoculum, was used to establish a baseline to compare with experimental treatments, and the system was in batch mode.

Next, the fermentation bottle that had been filled with the fermentable material was placed in a hot water bath at a temperature of 35 ± 1 °C to measure the volume of biogas produced daily. Some of the water was replaced with gas to analyze the gas composition using GC. The parameters measured in wastewater samples are according to [11] (Table S1).

### 2.4.2. Semicontinuous Experiments

From the batch experiment, the optimum ratio and conditions efficient in producing the maximum methane yield were selected. This involves studying the optimum storage time and the efficiency in treating organic matter, as well as the rate of methane production in the system. In this study, a semicontinuous CSTR with an airless system was used to control the experimental conditions suitable for anaerobic fermentation at a constant temperature of 35 ± 1 °C. The composition of the gas was analyzed using GC every seven days, and the obtained methane percentage was multiplied by the volume of biogas produced to calculate the amount of methane produced each day. The diagram of the semicontinuous digester is presented in Figure 2.

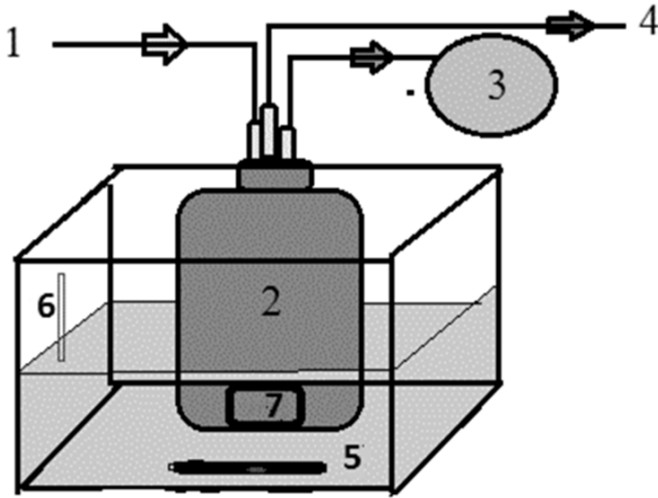

**Figure 2.** Diagram of semicontinuous digester including (1) inlet; (2) digester 4 L; (3) gas collector; (4) outlet; (5) heater bath control temperature for 35 ± 1 °C; (6) thermometer; and (7) stirrer.

The reactor performs experiments at different storage times or hydraulic retention times (HRTs) of 30, 25, 20, 14, and 7 days, consecutively. The longer HRT was preferred by the high buffer system. The short HRT is desirable in terms of minimizing the capital cost. The variation in storage times studied were in the range of the optimized anaerobic degradation rate from many studies [12–15]. A mixing rate of 100% means stirring for 24 h. The effluent discharge from the reactors was carried out once daily with volumes of 100, 120, 150, 214, and 428 mL, consecutively (Table 2). The volume of wastewater discharged depended on the liquid retention in the system. During the discharge, the wastewater was sucked through a feeding tube before refilling it with new wastewater. Continuous stirring was maintained during the discharge to ensure that the effluent had a consistent texture. Water samples were collected for chemical analysis (Table 3) to monitor the system performance by considering variable parameters, such as pH, alkalinity, COD, VFA, and MLSS. The biological gas produced was discharged in the balloon connected to the gas counter. The amount of biological gas produced daily was then measured. Gas samples were collected in a tube and placed in a vacuum test tube to analyze the gas composition. After measuring the gas, the gas line was closed. After completing the wastewater discharge, the effluent line was closed, and the wastewater line was opened to refill the system with the volume of discharged wastewater. The water inlet was then closed, and the balloon was opened to collect the biogas. The system was run until it reached a steady state, based on the volume and composition of biogas that varies within ±15%.

**Table 2.** Type of fermentation material, retention time, and volume of waste added and discharged daily in the semicontinuous system for biogas production experiments.

| Set No. | Type of Substrate | Working Volume (mL) | HRT (Day) | Waste Added–Discharged (mL) |
|---|---|---|---|---|
| 1 | POME + 50% seed + 5% EFB ww | 3000 | 7 | 428 |
| 2 | POME + 50% seed + 5% EFB ww | 3000 | 14 | 214 |
| 3 | POME + 50% seed + 5% EFB ww | 3000 | 20 | 150 |
| 4 | POME + 50% seed + 5% EFB ww | 3000 | 25 | 120 |
| 5 | POME + 50% seed + 5% EFB ww | 3000 | 30 | 100 |

Note: ww—wastewater.

**Table 3.** Measured parameters, analytical method, frequency sampling of anaerobic fermentation systems.

| Parameter | Method | Frequency of Monitoring |
|---|---|---|
| Total solids | Gravimetric method | every 3 days |
| COD | Close reflux, Titrimetric | every 3 day |
| Temperature | Thermometer | every day |
| pH | pH meter | every day |
| Alkalinity | Direct titration method | every 6 days |
| VFA | Direct titration method | every 6 days |
| Biogas production | fluid displacement | every day |
| Biogas composition | GC-TCD | every week |

### 2.4.3. Measuring the Volume of Biogas, Analyzing the Composition of the Gas and Liquid Samples

The biogas produced was measured in volumes using water displacement and collected for analysis. The biogas composition, including methane ($CH_4$) and carbon dioxide ($CO_2$), was analyzed every 12 days using GC with a thermal conductivity detector (TCD) sensor. The column used was an HP-plot/Q with a diameter of 1 mm and a length of 2 m, using helium gas as the carrier. The temperature in the injector, detector, and oven was maintained at 250 °C.

The characteristics of the influent and effluent of the anaerobic digestion system were analyzed according to Standard Methods for the Examination of Water and Wastewater, 22nd edition [11]. The parameters to be analyzed included total solids (TS), volatile solids (VS), volatile fatty acids (VFA), pH, alkalinity, and COD (Table 3).

2.4.4. Analysis of the Influent and Effluent Characteristics and Sludge from the Anaerobic Digestion System

This experiment used the residual sludge from the appropriate retention time of a semicontinuous anaerobic fermentation process. The analysis aimed to determine the amount of plant nutrients, including total nitrogen, total phosphorus, and potassium as well as the C/N ratio, moisture content, and organic carbon content. The results were used to apply the sludge as a soil conditioner material and were compared to the Thai Industrial Standard for organic compost set by the Industrial Product Standards Office, Ministry of Industry [16].

*2.5. Circular Economy Capacity Analysis*

The results obtained from this research experiment could be used as information to support decision making for entrepreneurs interested in producing electricity using the anaerobic wastewater treatment system of crude palm oil mills. The analysis focuses on identifying the costs, returns, and feasibility of using EFB wastewater in codigestion with POME to produce biogas as an energy source or fuel to generate electricity. Economic tools are used to evaluate the profitability and feasibility of the proposed system. This study refers to reliable secondary data and information obtained from supporting mills, which define the conditions for analysis as follows.

(1)   Setting assumptions for project analysis.

- The operational period of the biogas power plant is five years based on the minimum years of the power purchase agreement with the regional electricity authority, in a firm pricing scheme.
- The return on investment starts from year 1 and ends at the project's completion.
- No salvage value is considered at the end of the operational period.
- A discount rate of 10% is used.

(2)   Analyzing the cost of the biogas power plant can be categorized as follows.

(2.1)   The investment cost or fixed cost is the initial cost incurred from constructing the power plant and equipment, including the following:

- The cost of the biogas system, consisting of the biogas digester tank, gas storage tank, piping system, and monitoring and control equipment.
- The cost of the electricity production system, consisting of the biogas-powered electricity generator, control equipment, gas piping system, and electrical wiring.

(2.2)   The operating cost or variable cost is the expense for general management and production, including labor cost and operational and maintenance expenses, calculated on an annual basis.

(3)   Project Return Analysis: The business will generate revenue from selling electricity produced and selling it back to the state through the Small Power Producer (SPP) and Very SPP (VSPP) policies. This includes the value of the building and land when the project is completed. Machinery and equipment are considered to have zero salvage value at the end of their useful life.

## 3. Results and Discussion

*3.1. The Chemical Characteristics of Wastewater from Palm Oil Mill Extraction, Wastewater from EFB Pressing, and Sludge*

Both types of wastewater have different physical characteristics due to their different sources in the production process. Additionally, both types of wastewater have high levels

of pollution that can cause water contamination if released into the external environment, especially the wastewater from the EFB pressing. This waste is generated as an additional byproduct of the main production process. The process of EFB pressing uses heat to help release organic substances that remain in the EFB, resulting in a large amount of organic matter contaminating the wastewater.

Typically, the factory can treat the general wastewater from the crude palm oil extraction process directly with an anaerobic sequencing batch reactor and upflow anaerobic sludge blanket treatment systems producing biogas as a renewable energy source for electricity generation. However, the factory collects the wastewater from the EFB extraction for later treatment. The wastewater is fed into the treatment system slowly and gradually due to its high level of pollution, which could cause the treatment system to fail if too much wastewater is added at once.

The results of the chemical analysis of two types of wastewater from the palm oil extraction process and sludge showed high levels of pollution in both types, as indicated by their COD and BOD values. The general wastewater from the palm oil extraction process had COD and BOD values of 61,000 and 29,798 milligrams per liter, respectively, while the wastewater from the EFB extraction had COD and BOD values of 74,750 and 31,339 mg/L, respectively. When comparing the pollution levels of both types of wastewater, the wastewater from the EFB of oil palm had higher levels of pollution than the general wastewater from the palm oil extraction process (Table 4).

**Table 4.** Chemical characteristics of wastewater and sludge used in the experiment.

| Parameter | POME | EFB Wastewater | Sludge |
|---|---|---|---|
| pH | 4.6 | 4.9 | - |
| COD (mg/L) | 61,000 | 74,750 | - |
| BOD (mg/L) | 29,798 | 31,339 | - |
| TKN (mg/L) | 550 | 325 | - |
| $NH_3$-N (mg/L) | 2.75 | 5.25 | - |
| Grease and oil (mg/L) | 970 | 8590 | - |
| TS (mg/L) | 20,010 | 96,320 | - |
| SS (mg/L) | 16,250 | 91,240 | - |
| VFA (mg/L) | 5288 | 10,613 | - |
| MLSS (mg/L) | 18,000 | - | 7.63 |
| BOD:COD | 0.48 | 0.41 | |

Based on these properties, if methane production is considered, 1 g of degraded COD will produce 0.351 L of methane. This means that the wastewater from the palm oil extraction process can produce 21.41 L of methane, while the wastewater from the EFB extraction can produce 26.24 L of methane. This theory shows that the wastewater from EFB pressing can produce more methane than the general wastewater from the palm oil extraction process, despite both types of wastewater having high levels of organic matter. This is because both types of wastewater can be a sufficient source of methane when treated in anaerobic conditions due to their high levels of easily biodegradable organic matter, as evidenced by their BOD/COD ratios ranging from 0.4 to 0.5 [17,18] (Table 4).

Apart from high levels of organic compounds in the form of BOD or COD, both types of wastewater also contain high amounts of volatile fatty acids. These fatty acids are considered organic pollutants in wastewater. However, they can be converted to biogas, although the process of converting fat substrate into biogas is difficult [5]. Simply adding only empty palm oil fruit bunches pressing wastewater in the anaerobic digestion process to produce methane may result in system failure due to the high levels of impurities and volatile fatty acids. This can cause volatile fatty acids to accumulate, which can be toxic to the group of microorganisms producing methane, leading to inhibition of the methane production process. Therefore, codigestion with palm oil mill effluent can dilute the toxicity that may occur in the system and adjust the nutrient ratio to be suitable for efficient microbial work, resulting in increased biodegradation rates and biogas production [19].

The black-colored microorganisms in the sludge have a high SS concentration. Regarding the concentration of microorganisms in the reactor tank, the MLSS is 18,000 mg/L, and the pH value is 7.63. The sludge was analyzed and used as inoculum in the fermentation system. The inoculum was taken directly from the biogas production tank of the factory, making it suitable for use as an inoculum in the fermentation process, which was consistent with other research [5,6,20]. The potential for generating gas from palm oil mill effluent was studied using a 2 L CSTR reactor with a stirring speed of 100 rounds per minute, operating at a thermophilic state and temperature of 55 °C for 6 days. The study found that at an MLSS concentration of 14,000 mg/L, the COD removal efficiency reached 90%, with a methane content of 64%.

### 3.2. The Study of Codigestion of POME and EFB Wastewater at Various Ratios Using a Batch System

The results of a batch codigestion process, using different mixing ratios and lasting ten days until the end of the fermentation process, are presented in Figure 3. Experimental set 7, comprising a mixing ratio of 45% POME + 50% seed + 5% EFB wastewater, exhibited the highest cumulative biogas and methane yields of 396 ± 4.58 mL and 294 ± 3.51 mL, respectively, compared with the other 11 experimental sets.

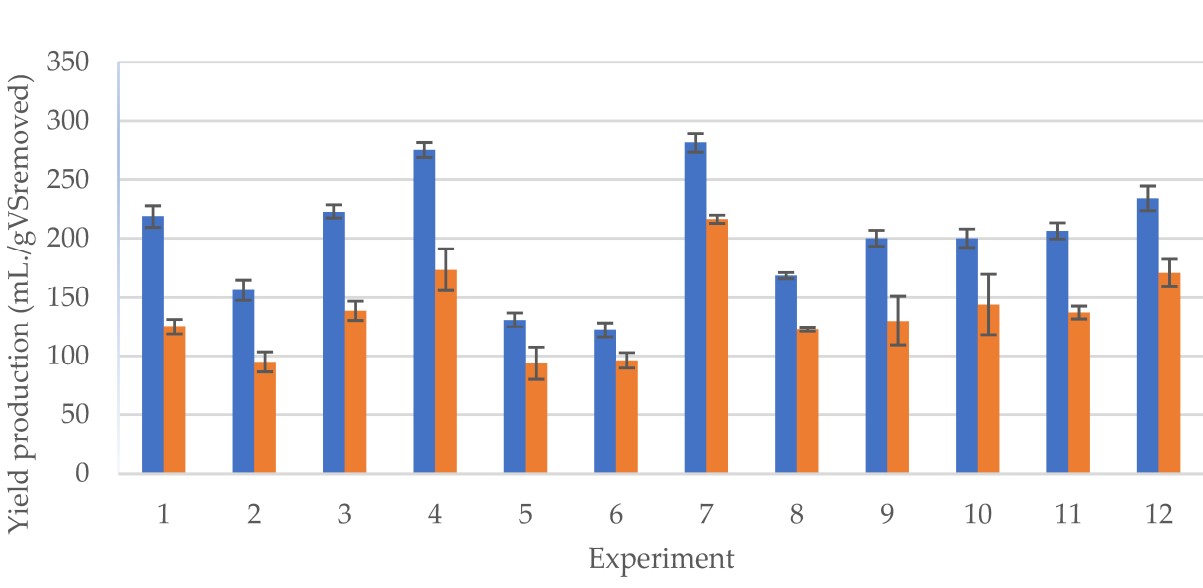

**Figure 3.** Biogas and methane yield for each experiment.

This resulted in the highest methane production efficiency of 0.016 L $CH_4$/g VS removed or 0.18 L $CH_4$/g VS added, increasing when cofermented with EFB wastewater, compared with the fermentation of POME and inoculum alone, as shown in Figure 4.

In Table 5, the biogas production is expressed in units of L $CH_4$/g VS removed to show the amount of methane produced from the organic matter decomposed in the substrate and in units of L $CH_4$/g VS added to show the total methane produced from all substrates, including the portion that cannot be decomposed by microorganisms.

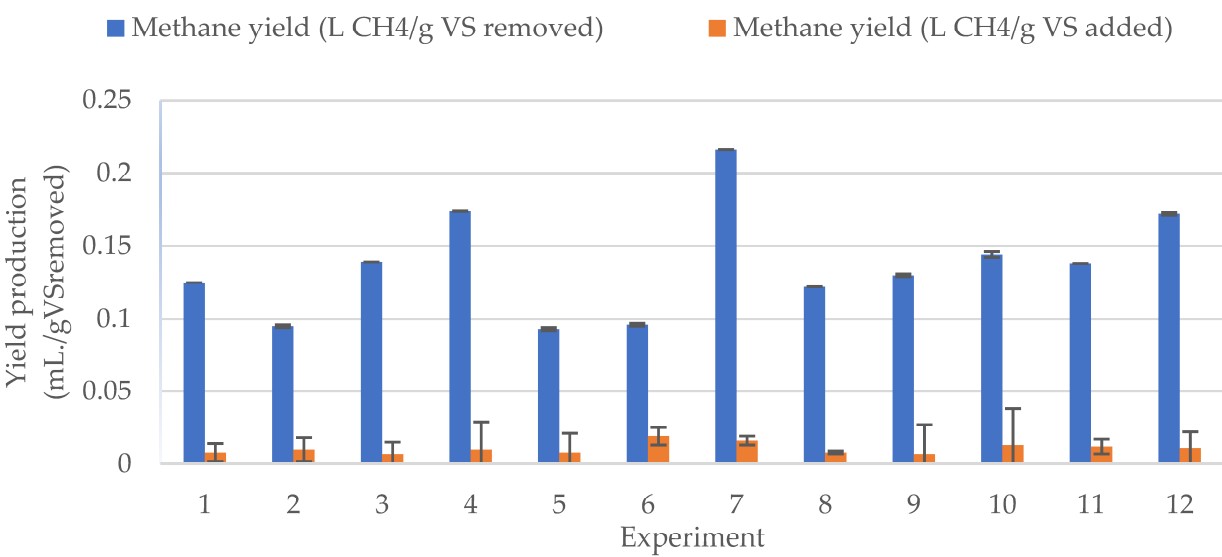

**Figure 4.** Methane yield by VS removal and VS added of each experiment.

**Table 5.** Biomethane potential of batch experiments.

| No. | Condition (Total Volume 100%) | mL CH$_4$ | VS Added (g/L) | VS Removed (g/L) | Methane Yield | |
|---|---|---|---|---|---|---|
| | | | | | L CH$_4$/gVS Removed | L CH$_4$/gVS Added |
| 1 | POME + 35% seed (control) | 124 ± 6.43 | 16.12 | 0.99 | 0.125 ± 0.006 | 0.008 ± 0.000 |
| 2 | POME + 35% seed + 2.5% EFB ww | 211 ± 18.52 | 20.28 | 2.22 | 0.095 ± 0.008 | 0.010 ± 0.001 |
| 3 | POME + 35% seed + 5% EFB ww | 150 ± 9.02 | 22.41 | 1.08 | 0.139 ± 0.008 | 0.007 ± 0.000 |
| 4 | POME + 35% seed + 10% EFB ww | 170 ± 17.62 | 17.26 | 0.98 | 0.174 ± 0.018 | 0.010 ± 0.001 |
| 5 | POME + 50% seed (control) | 157 ± 21.36 | 15.98 | 1.67 | 0.093 ± 0.013 | 0.008 ± 0.001 |
| 6 | POME + 50% seed + 2.5% EFB ww | 233 ± 14.64 | 19.73 | 2.42 | 0.096 ± 0.006 | 0.019 ± 0.001 |
| 7 | POME + 50% seed + 5% EFB ww | 294 ± 3.51 | 18.46 | 1.36 | 0.216 ± 0.003 | 0.016 ± 0.000 |
| 8 | POME + 50% seed + 10% EFB ww | 140 ± 1.53 | 16.59 | 1.14 | 0.122 ± 0.001 | 0.008 ± 0.000 |
| 9 | POME + 75% seed (control) | 138 ± 15.72 | 18.99 | 1.06 | 0.130 ± 0.020 | 0.007 ± 0.001 |
| 10 | POME + 75% seed + 2.5% EFB ww | 229 ± 39.89 | 17.41 | 1.59 | 0.144 ± 0.025 | 0.013 ± 0.002 |
| 11 | POME + 75% seed + 5% EFB ww | 209 ± 8.33 | 17.61 | 1.52 | 0.138 ± 0.005 | 0.012 ± 0.000 |
| 12 | POME + 75% seed + 10% EFB ww | 212 ± 13.20 | 19.87 | 1.24 | 0.172 ± 0.011 | 0.011 ± 0.001 |

Note: ww—wastewater.

Experimental set 7 exhibited the highest methane production efficiency, occurring during codigestion of POME + 50% seed + 5% EFB wastewater, with a value of 0.216 ± 0.003 L CH4/g VS removed or 0.0159 ± 0.000 L CH$_4$/g VS added. This showed that codigestion with EFB wastewater could increase the methane production efficiency compared with the control set (experiment set 3), revealing codigestion of 50% POME + 50% seed without EFB wastewater, with a value of 0.093 ± 0.013 L CH$_4$/g VS removed or 0.0078 ± 0.001 L CH$_4$/g VS added. The results demonstrate that codigestion could increase the methane production efficiency up to twofold (Figures 3 and 4), which was consistent with other studies on codigestion to increase the methane production efficiency [9,19,21,22]

Upon analyzing the default parameters listed in Table 6, decreasing the volume of POME resulted in a reduced amount of initial volatile fatty acid (VFA) while increasing the quantity of inoculum. Typically, the recommended range for VFA is between 50 and 500 mg per liter of CH3COOH, with the maximum permissible value of 2000 mg per liter of CH3COOH in the system [23]. However, the initial VFA quantity in sets 1 to 4 exceeded the proposed theoretical value, rendering the conditions unfavorable for the gas production process. Moreover, the initial alkalinity levels in sets 2, 3, 4, and 7 exceeded 5000 mg per liter, whereas the acceptable range of general alkalinity is between 1000 and 5000 mg per

liter as calcium carbonate [17]. Nonetheless, upon analyzing the ratio of VFA to bicarbonate alkalinity (VFA/HCO$_3$), all the experimental sets had values lower than 0.4, suggesting that the system was buffered.

**Table 6.** Parameters related to before (influent) and after (effluent) digestion processes, initial inoculum at 35%.

| Parameter | 65% POME + 35% Seed (Control) | | 62.5% POME + 35% Seed +2.5% EFB ww | | 60% POME + 35% Seed +5% EFB ww | | 55% POME + 35% Seed +10% EFB ww | |
|---|---|---|---|---|---|---|---|---|
| | Influent | Effluent | Influent | Effluent | Influent | Effluent | Influent | Effluent |
| pH | 7.07 | 8.30 | 7.04 | 8.66 | 7.04 | 8.67 | 7.07 | 8.77 |
| COD, mg/L | 43,444 | 42,792 | 43,164 | 34,639 | 49,780 | 31,838 | 63,383 | 57,373 |
| SS, mg/L | 26,000 | 16,830 | 28,870 | 17,450 | 24,640 | 14,120 | 25,850 | 17,480 |
| TS, mg/L | 45,384 | 29,788 | 37,516 | 35,544 | 38,584 | 33,600 | 50,704 | 38,728 |
| VS, mg/L | 16,124 | 15,132 | 20,228 | 18,004 | 22,412 | 17,336 | 17,256 | 16,280 |
| Alkalinity, mg/L | 4800 | 5000 | 6400 | 8600 | 5600 | 7000 | 6600 | 8200 |
| VFA, mg CaCO$_3$/L | 1900 | 680 | 2160 | 400 | 2280 | 640 | 2300 | 1160 |
| NH$_3$-N, mg/L | 169 | 260 | 125 | 265 | 180 | 295 | 124 | 255 |
| TKN, mg/L | 309 | 465 | 304 | 464 | 368 | 445 | 305 | 432 |
| C:N | 140:1 | 92:1 | 142:1 | 74:1 | 135:1 | 71:1 | 208:1 | 133:1 |

Note: ww—wastewater.

From Tables 6–8, the effluents were lower than the influents, indicating that the organic matter in the system was being degraded into gas. Furthermore, the pH, alkalinity, NH$_3$-N, and TKN of the effluent increased. The fermentation process ended due to an increase in NH$_4$ during the final stage, leading to a pH value higher than 8, indicating an imbalance in the system's metabolism that can be toxic to the group of methane-producing bacteria. In general, the appropriate conditions for this group of bacteria should have a pH value within the range of 7 to 8 [24,25].

**Table 7.** Parameters related to before (influent) and after (effluent) digestion processes, initial inoculum at 50%.

| Parameter | POME + 50% Seed (Control) | | POME + 50% Seed +2.5% EFB ww | | POME + 50% Seed +5% EFB ww | | POME + 50% Seed +10% EFB ww | |
|---|---|---|---|---|---|---|---|---|
| | Influent | Effluent | Influent | Effluent | Influent | Effluent | Influent | Effluent |
| pH | 7.05 | 8.54 | 7.07 | 8.68 | 7.06 | 8.55 | 7.06 | 8.55 |
| COD, mg/L | 50,245 | 28,038 | 62,110 | 33,630 | 63,608 | 31,600 | 65,919 | 30,588 |
| SS, mg/L | 29,120 | 24,060 | 29,360 | 24,620 | 27,820 | 37,310 | 55,010 | 49,720 |
| TS, mg/L | 46,388 | 32,412 | 48,120 | 44,284 | 44,968 | 39,408 | 39,516 | 36,228 |
| VS, mg/L | 15,180 | 14,296 | 19,728 | 17,312 | 18,460 | 17,620 | 16,588 | 15,444 |
| Alkalinity, mg/L | 4800 | 5500 | 4800 | 6200 | 5750 | 4750 | 4750 | 5500 |
| VFA, mg CaCO$_3$/L | 1260 | 250 | 1300 | 380 | 1200 | 325 | 1100 | 300 |
| NH$_3$-N, mg/L | 240 | 355 | 227 | 338 | 231 | 336 | 221 | 312 |
| TKN, mg/L | 794 | 1660 | 533 | 616 | 504 | 672 | 448 | 616 |
| C:N | 63:1 | 17:1 | 116:1 | 54:1 | 126:1 | 47:1 | 147:1 | 50:1 |

Note: ww—wastewater.

*3.3. Study of Codigestion of POME and EFB Wastewater with Various Retention Times Using a Semicontinuous System*

The semicontinuous fermentation process was carried out using a mixture ratio of 45% POME, 50% seed, and 5% EFB wastewater. This ratio was used for the initial fermentation and allowed to be retained for 10 to 30 days.

**Table 8.** Parameters related to before (influent) and after (effluent) digestion processes, initial inoculum at 75%.

| Parameter | POME + 75% Seed (Control) | | POME + 75% Seed +2.5% EFB ww | | POME + 75% Seed +5% EFB ww | | POME + 75% Seed +10% EFB ww | |
|---|---|---|---|---|---|---|---|---|
| | Influent | Effluent | Influent | Effluent | Influent | Effluent | Influent | Effluent |
| pH | 7.09 | 8.79 | 7.09 | 8.68 | 7.05 | 8.64 | 7.07 | 8.59 |
| COD, mg/L | 58,082 | 25,034 | 62,340 | 22,813 | 65,741 | 27,478 | 74,903 | 36,947 |
| SS, mg/L | 37,520 | 32,370 | 41,960 | 35,330 | 59,260 | 28,490 | 66,050 | 32,470 |
| TS, mg/L | 48,792 | 44,280 | 51,476 | 44,528 | 49,660 | 45,980 | 49,720 | 44,524 |
| VS, mg/L | 18,992 | 16,648 | 17,008 | 16,500 | 17,212 | 16,496 | 19,872 | 18,836 |
| Alkalinity, mg/L | 3750 | 4750 | 4750 | 5000 | 4500 | 4500 | 4400 | 5200 |
| VFA, mg $CaCO_3$/L | 1425 | 450 | 850 | 200 | 525 | 250 | 500 | 240 |
| $NH_3$-N, mg/L | 252 | 266 | 289 | 390 | 305 | 395 | 315 | 379 |
| TKN, mg/L | 784 | 840 | 672 | 728 | 560 | 616 | 952 | 1568 |
| C:N | 74:1 | 30:1 | 93:1 | 31:1 | 117:1 | 46:1 | 78:1 | 23:1 |

Note: ww—wastewater.

### 3.3.1. Effect of the Retention Time on Methane Production Rate and Methane Yield

According to the study, the cofermentation of 45% POME, 50% seed, and 5% EFB wastewater resulted in the accumulation of biogas and methane. The accumulated biogas volumes after 7, 14, 20, 25, and 30 days of retention time were 33,963, 46,870, 45,841, 67,558, and 65,868 mL, respectively (as shown in Figure 5). The accumulated methane volumes after 7, 14, 20, 25, and 30 days of retention time were 1685, 9029, 10,549, 28,470, and 23,793 mL, respectively. At a retention time of 25 days, the organic loading rate (OLR) was 2.60 g COD/L·day, and the accumulated biogas and methane volumes were higher than those at other retention times during the entire 30-day experiment.

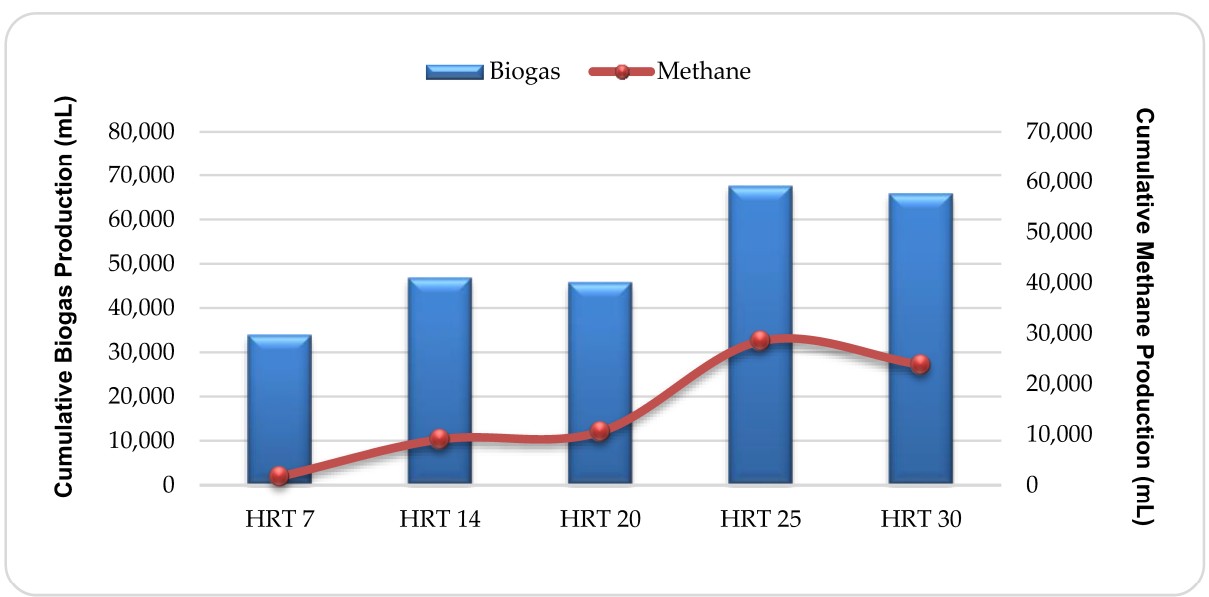

**Figure 5.** Accumulated biogas and methane volumes at the retention time of 30 days from the semicontinuous fermentation process.

Gas production with methane generated each day (illustrated in Figure 6) was found to have low methane production during the short retention times of 7 and 14 days, with an OLR of 9.27 g COD/L·day and 4.63 g COD/L·day, respectively. This was due to system failure caused by an increased amount of COD from adding wastewater to the system. Methane production stopped at 17 and 19 days, respectively. At a retention time of 20 days with an OLR of 3.25 g COD/L·day, methane production was observed between days 4 and 20 and stopped on day 21. Across those three retention times, the maximum methane production was only 7, 32, and 39%, respectively, indicating that these retention times

were unsuitable for methane production. At retention times of 25 and 30 days, methane production was as high as 62 and 53%, respectively, indicating that these retention times were suitable for methane production. The system reached a steady state on days 25 and 22, respectively, and the fermentation process was completed after 30 and 27 days, respectively.

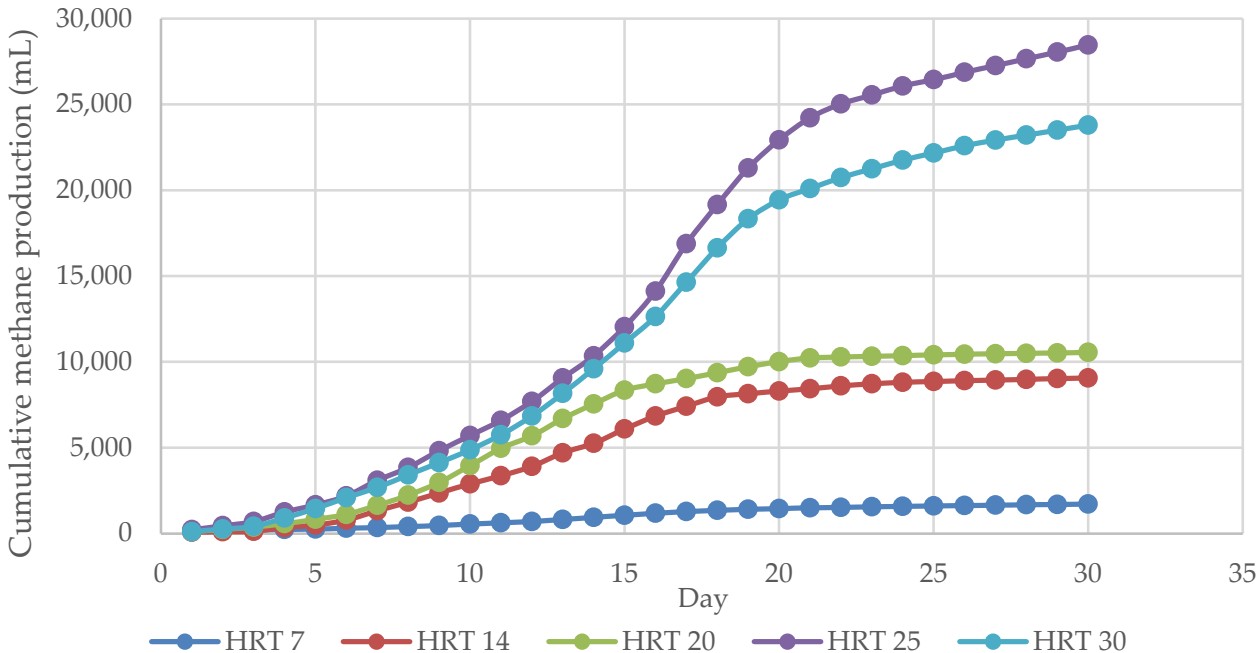

**Figure 6.** Amount of methane produced each day at different retention times using a semicontinuous system.

The results of the experiment are consistent with [26] study on the codigestion of wastewater and solid waste from an olive oil production plant in a mesophilic condition, with an organic loading rate ranging from 0.67 to 6.67 gCOD/L·day. The study found that biogas production was inhibited when the organic loading rate exceeded 4.67 gCOD/L·day.

Experiments conducted with a wide range of HRTs have proven beneficial for manufacturing operations in subsequent up-scale experiments in the biogas system. The variation in the HRT corresponds to changes in the organic loading rate (OLR), which can be traced back to variations in chemical oxygen demand (COD). Manufacturers can further refine the feeding conditions of the biogas system by calculating the OLR based on the instantaneous COD of the substrate, thus maximizing biogas production.

3.3.2. Changes in the System during Different Retention Times in Semicontinuous Anaerobic Fermentation Process

During the cofermentation process of POME + 50% seed + 5% EFB in a semicontinuous mode at different retention times, at a retention time of 7 days, the pH value (illustrated in Figure 7) decreased rapidly due to the accumulation of volatile fatty acids (VFAs) within the system. At retention times of 14 and 20 days, during the first 1 to 3 days of operation, the pH value decreased, but then gradually increased, even though it remained within the suitable range. However, the high organic loading and short retention time resulted in low methane production and a rapid end to the fermentation process.

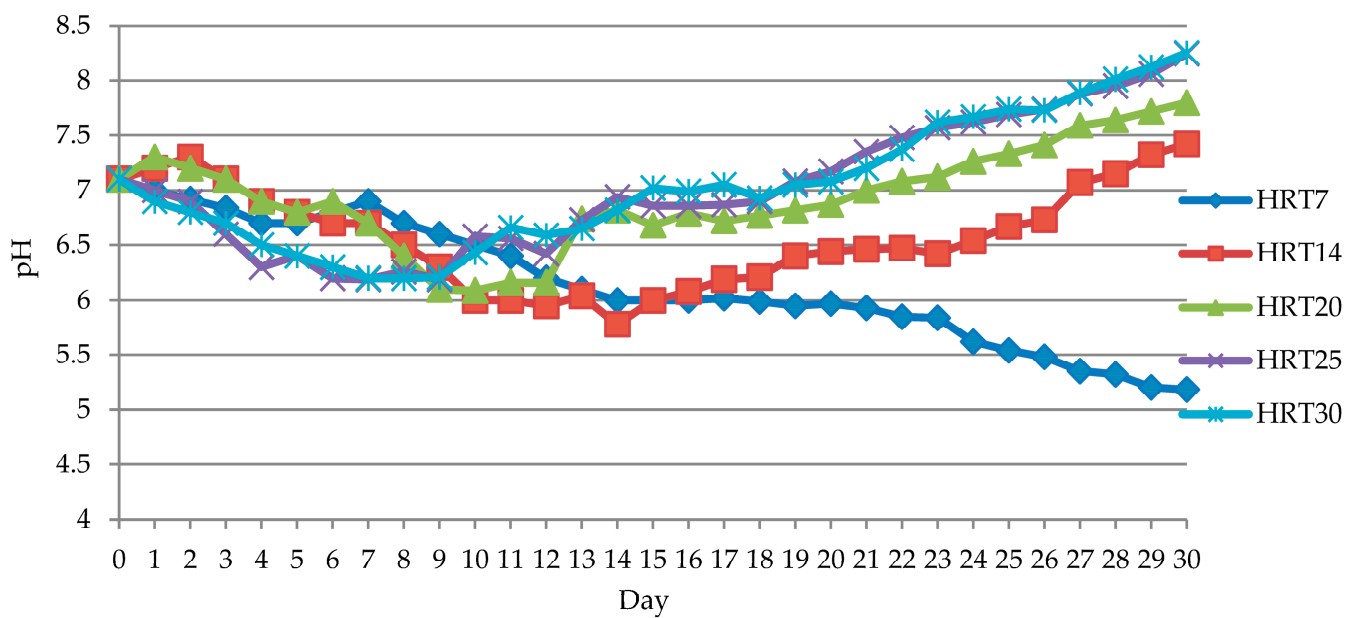

**Figure 7.** Changes in pH values at different retention times.

At the retention times of 25 and 30 days, the pH value continuously increased and remained within the suitable range, resulting in a higher methane production compared with other retention times. This indicated that a longer retention time was more efficient than a shorter one, especially for systems with high nutrient or COD concentrations. In the case of retention times of 25 and 30 days, the process could run up to 30 and 28 days, respectively. However, if the pH value exceeded 8.0, the fermentation process would end.

Nutrients containing fats and long-chain fatty acids (LCFAs) often lead to easy inhibition of the fermentation process. LCFAs can affect the methanogenesis process and other steps in anaerobic digestion. As shown in Figure 8, VFA accumulation increased at retention times of 7, 14, 20, and 25 days due to the degradation of organic matter under anaerobic conditions. However, at a retention time of 30 days, the VFA levels were slightly increased in the beginning and decreased later than 18 days, indicating the consumption of VFAs by microorganisms to produce methane gas and insufficient organic matter in the later phase (longer HRT provided lower OLR). The alkalinity values continuously decreased throughout all the retention times, affecting the buffer capacity of the reactor and ultimately leading to the end of the fermentation process.

### 3.3.3. COD Removal Efficiency

After the fermentation process experiment, at retention times of 25 and 30 days, the efficiency of the COD removal was up to 67 and 84%, respectively (Table 9). This was because longer retention times allow microorganisms to use nutrients for a longer period. In contrast, shorter retention times reduced the efficiency of COD removal. Specifically, at a retention time of 20 days, the efficiency of COD removal was only 45%, and at retention times of 7 and 14 days, no efficiency was observed in COD removal due to the short retention time. This was consistent with [26] research on the anaerobic codigestion of industrial wastewater and solid waste from an olive oil factory. The experiment was conducted at an average temperature, with retention times of 12, 24, and 36 days, and the wastewater had COD concentrations of 24, 56, and 80 gCOD/L. The highest organic removal efficiency was 89%, achieved with an organic loading rate of 0.67 gCOD/L/day (with a wastewater concentration of 24 gCOD/L) and a retention time of 36 days.

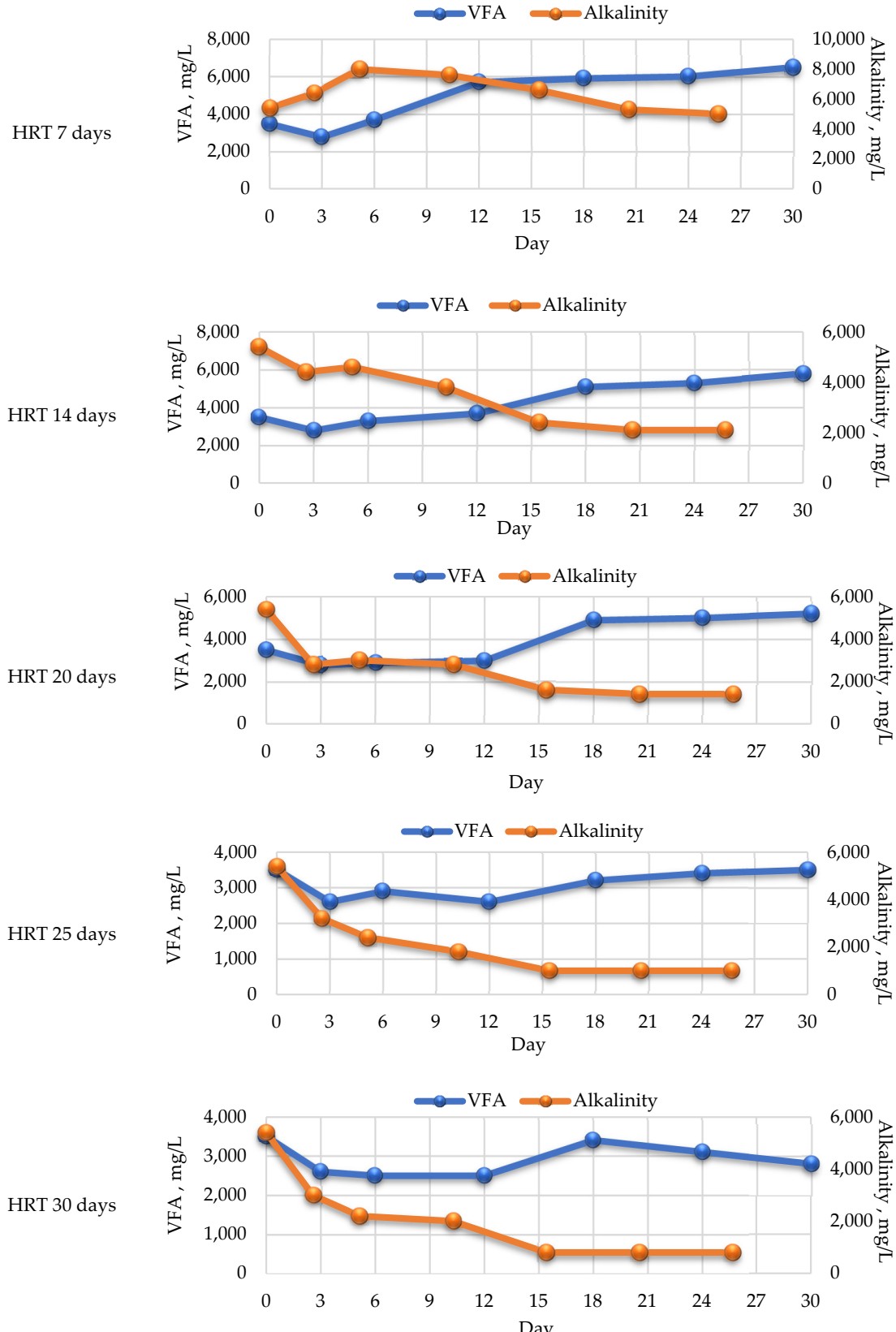

**Figure 8.** VFA and alkalinity of the experiments with different HRTs.

**Table 9.** Effluent characteristics of different retention time experiments.

| Parameter | Influent | Effluent | | | | |
|---|---|---|---|---|---|---|
| | | HRT7 | HRT14 | HRT20 | HRT25 | HRT30 |
| pH | 7.1 | 5.18 | 7.42 | 7.80 | 8.24 | 8.25 |
| COD (mg/L) | 96,000 | 128,000 | 112,000 | 52,800 | 32,000 | 16,000 |
| TKN (mg/L) | 392 | 252 | 560 | 476 | 588 | 504 |
| $NH_3$-N (mg/L) | 252 | 280 | 224 | 280 | 392 | 392 |
| TS (mg/L) | 33,172 | 29,756 | 20,748 | 28,632 | 29,572 | 19,332 |
| SS (mg/L) | 28,305 | 17,703 | 17,790 | 11,616 | 20,686 | 19,508 |
| VFA (mg/L) | 3500 | 5900 | 5100 | 4900 | 3400 | 3100 |
| Alkalinity (mg/L) | 5400 | 6600 | 2400 | 1600 | 1000 | 800 |
| C:N | 131:1 | 106:1 | 92:1 | 102:1 | 75:1 | 49:1 |
| COD removal efficiency (%) | | −34 | −17 | 45 | 67 | 84 |

When considering all the consensus results obtained from different HRTs, the 25 days was a suitable HRT with the highest cumulative biogas and methane production, the continuous VFA production during the experimental period contributing to methanogenesis continuously.

3.3.4. Characteristics of Remaining Sludge after Fermentation

An analysis of the characteristics of the sludge residue included the purpose of recycling waste for further use, such as soil conditioner, making fertilizer for plants, and so on. The sludge analyzed (Figure S1) was taken from the fermentation system with a suitable retention period of 25 days, providing the highest amount of nitrogen compared with other retention periods.

The properties of the sludge residue (Table 10) have an organic carbon value of 19.77%, total nitrogen value (N) of 2.62%, total phosphorus value (Total $P_2O_5$) of 2.80%, and total potassium value ($K_2O$) of 11.32%. The ratio of carbon to nitrogen was 7.0, whereas the moisture content value was 28.55%. These values fall within the standard range issued by the Department of Agriculture, Ministry of Agriculture and Cooperatives, Thailand [27].

**Table 10.** Sludge residue characteristics.

| Parameter | % *w/w* | Standard |
|---|---|---|
| Organic carbon | 19.77 | Not lower than 20 |
| Total N | 2.62 | Not lower than 1.00 |
| Total $P_2O_5$ | 2.80 | Not lower than 0.5 |
| Total $K_2O$ | 11.32 | Not lower than 0.5 |
| C/N ratio | 7.00 | Not exceeding 20:1 |
| Moisture content | 28.55 | |

*3.4. Circular Economy Perspective through Economic Analysis*

Data from the factory supported research found that the factory had incurred expenses in setting up a biogas production system equal to 2,625,889 USD, including the construction cost, maintenance cost, electricity cost, and chemicals and materials (see Supplementary Materials Table S2).

During the production of crude palm oil, the factory generates EFBs at a rate of approximately 25% of the fresh fruit bunches. With a current production rate of 720 tons of fresh fruit bunches daily, this results in around 180 tons of EFBs daily. To increase production efficiency, a crude palm oil company has implemented a process of compressing the EFBs, as there is still residual oil in the EFBs that can be extracted and reused. This process increases oil production and helps to reduce oil pollution in the environment. With approximately 180 tons of EFBs daily, the EFB compression process generates wastewater at a rate of 54 cubic meters daily or $1.6 \times 10^4$ cubic meters yearly. Additionally, 1 kg of EFB

can be converted to 0.00722 kg of oil, and the cost of adding the EFB compression process to the production process was 148,885 USD (Table S3).

From the codigestion experiment, the appropriate ratio of general wastewater from the palm oil extraction factory to EFBs was 45% POME + 50% seed + 5% EFB wastewater. When a working volume of 0.5 L was used with the above ratio, 0.294 L of methane gas was produced, which was greater than the control set (without EFB wastewater), producing only 0.157 L of methane gas. This showed the potential for methane gas production to increase up to twice when using EFB wastewater of 0.0125 L mixed with POME 0.238 L. When the EFB wastewater produced was $1.6 \times 10^7$ L/year, it required POME $3.0 \times 10^8$ L/year.

Therefore, the total volume of the two materials fermented together was $3.16 \times 10^8$ L/year. When the above ratio was tested in a semicontinuous CSTR reactor system, 0.95 L/day of methane gas was produced with an organic feed rate of 0.15 L/day. When the volume of the fermented material was equal to $3.16 \times 10^8$ L/year, methane gas could be produced at a rate of $2.0 \times 10^6$ m$^3$/year.

The results of the experiment showed that mixing EFB wastewater with POME could increase the production of biogas. If this project is implemented, the cost can be calculated based on the information presented in Table S4. Project A will produce biogas solely from the POME, requiring 2,625,889 USD, while Project B will produce biogas by codigestion of EFB wastewater with POME, requiring 2,804,431 USD.

*3.5. Economic Feasibility Assessment*

From the data provided by the supported factory, the yearly production of biogas was 3,600,000 cubic meters, which could be converted to 2.2 kWh of electricity per cubic meter of biogas. As a result, if the factory supplies this electricity to the power grid at a rate of 0.12 USD per unit, it could generate revenue of 939,502 USD yearly (Table S5). Moreover, by combining EFB wastewater with POME, the factory can double the amount of biogas produced. Therefore, the factory can generate electricity of $15.84 \times 10^6$ kWh per year, resulting in revenue of 1,879,004 USD yearly, while also increasing crude oil production of 388 tons/year or 345,907 USD/year. The preliminary data could be used to assist in making decisions about project construction using tools or criteria to evaluate the economic value. The economic viability of the project can be evaluated using the principles of a cost–benefit analysis, which consider the net present value and internal rate of return from Table S6. Both projects have constant revenue yearly and a project period of 5 years, with a minimum required return rate of 10%. From the table, it becomes evident that Project B, combining EFB wastewater with POME, had a higher net present value than Project A, with a net present value of 8,434,163 USD. When considering the internal rate of return of the projects, Project B had an internal rate of return of up to 73%, which was higher than Project A with an internal rate of return of 23%. Project A had a payback period of approximately 3 years, while Project B would have a payback period ranging from 1 to 2 years.

**4. Conclusions**

From the study results, the accumulation of biogas and methane occurred in cofermentation of the 45% POME + 50% seed + 5% EFB wastewater mixture during the retention periods of 30, 25, 20, 24, and 7 days, with cumulative biogas amounts of 15,024, 18,679, 11,896, 8120, and 6974 mL, respectively, and cumulative methane amounts of 4893, 6778, 2946, 1210, and 251 mL, respectively. The retention period of 25 days had higher cumulative biogas and methane amounts than the other retention periods throughout the entire experiment of 30 days. The characteristics of the sludge residue showed an organic carbon content of 19.17%, total nitrogen of 2.62%, total phosphorus of 2.80%, total potassium of 11.32%, a carbon-to-nitrogen ratio of 7.0, and a moisture content of 28.55%, showing its applicability for further use as organic compost. Based on the economic evaluation, it could be concluded that the codigestion of EFB wastewater with the 45% POME + 50% seed + 5% EFB wastewater mixture in the crude palm oil mill has a high economic value

when considering the net present value and internal rate of return, which are 8,434,163 USD and 73%, respectively.

**Supplementary Materials:** The following supporting information can be downloaded at: https://www.mdpi.com/article/10.3390/w15122153/s1, Table S1: Parameters used to measure and the method of analyzing the wastewater samples in the batch experiment. Table S2: The Biogas system installation cost. Table S3: The additional cost from EFB pressing installation. Table S4: The cost of EFB pressing installation into the system. Table S5. Income from projects. Table S6: Net Present Value (NPV) and Internal Rate of Return (IRR) from Projects. Figure S1: Sludge residue after 25 days of digestion.

**Author Contributions:** Conceptualization, T.T.S.; Validation, C.S.; Resources, N.C.; Data curation, K.J.; Writing—original draft, C.S.; Writing—review & editing, C.R. and T.T.S.; Project administration, T.T.S.; Funding acquisition, T.T.S. All authors have read and agreed to the published version of the manuscript.

**Funding:** This research was funded by Thailand Science Research and Innovation (TSRI; Grant No. MDS56I0177).

**Data Availability Statement:** No new data were created or analyzed in this study. Data sharing is not applicable to this article.

**Acknowledgments:** The authors are grateful to Thailand Science Research and Innovation (TSRI; Grant No. MDS56I0177) for their financial support.

**Conflicts of Interest:** The authors declare they have no conflict of interest, and the funders had no role in the design of the study; in the collection, analyses, and interpretation of the data; in the writing of the manuscript; or in the decision to publish the results.

## Abbreviations

| | |
|---|---|
| ASBR | anaerobic sequencing batch reactor |
| BOD | biochemical oxygen demand |
| COD | chemical oxygen demand |
| CSTR | semicontinuous completely stirred tank reactor |
| EFB | empty fruit bunch |
| G&O | grease and oil |
| HRT | hydraulic retention time |
| MLSS | mixed liquor suspended solids |
| OLR | organic loading rate |
| OPF | oil palm fronds |
| OPT | oil palm trunks |
| PPF | palm pressed fibers |
| POME | palm oil mill effluent |
| SS | suspended solids |
| TKN | total Kjeldahl nitrogen |
| TS | total solids |
| UASB | upflow anaerobic sludge blanket |
| VS | volatile solids |
| VFA | volatile fatty acids |

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
