# Peer review of "Promoting Circular Economy in the Palm Oil Industry through Biogas Codigestion of Palm Oil Mill Effluent and Empty Fruit Bunch Pressed Wastewater"

_water, doi:10.3390/w15122153_

Round 1

Reviewer 1 Report

This article described using wastewater in palm oil processing. The process is well presented and the result is appealing. The paper comes from an economic perspective which is very valuable. Recommend for publishing. 

Author Response

Dear Reviewer, We would like to express our sincere appreciation for you time and effort you dedicated to review our work and provide the positive comment. It is a great value for our team. Hopefully our work provides instructive point for further application.

Yours sincerely,

Reviewer 2 Report

The article is topical and considering the amounts of palm oil production - dealing with a significant applied problem. The study is based on a large amount of experimental work and will promote suggested way of wastewater treatment. Article provides a lot of experimental details, but rather design and upscaling aspects should be analysed.

Some following suggestions:

1. More upscaling possibilities should be discussed as well as transfer of lab scale process to continuos process

2. Cost calculation could be added as supplementary material

3. Writing of formulas should be checked. Presently m3, CH4 etc can be found

4. Page 4 more details on GC analysis could be provided

5. Writing of L should be checked. In Figures units should be L

6. List of abbreviations could be added

Moderate revision could be done

Reviewer 3 Report

This research work aims to investigate the biogas production and circular economy perspective in the palm oil industry through co-digestion of oil palm empty fruit bunch pressing wastewater and palm oil mill effluent, which is an interesting topic.  However, in my opinion, the introduction should detail more this problem. It´s only a problem in Thailand or it´s also a problem in other countries? Please add more references.

Please, through the text, try to use fewer acronyms, because difficult reading of the manuscript. 

During the manuscript, the future is used to describe the experiments that were done. I thing that the past simple should be used.

Line 98 - the authors referred "based on research studies", please mention the studies (references).

Line 156: explain the meaning of the sentence.

Table 3 - Column "Volume (ml)" of what?

line 244 - Add the title of subsection 3.1.1

Line 254 - add the meaning of ASBR and UASB

Add Table 5 after line 279

 Line 296 - "The researcher analyzed", which researcher, the author of this work or other researchers?

Please add error bars to the graphics of Figures 3, 4, 5, 6, 7, and 8. Replicates of the experiments were done? How many?

In Figures 3 and 4 where is represented the controller that is mentioned in line 334?

Line 337 - The authors mentioned "up to 2-fold", but this increase could be seen in which figure? How the value was obtained?

Line 368 - I didn't see in Table 7-9 the decrease that the authors refer to. Please explain better. 

Line 379-381 - Why were these the selected conditions?

Please pay attention to the section and subsection formation.

The graphs from Figure 8 are weird. The legend needs to be more detailed.

Figure 9 could go for supporting information. In my opinion, some tables and graphs/figures could also go for supporting information. 

Please add more references to the introduction and discussion.

There isn't a statistical analysis of the obtained data.

Need to be improved

Round 2

Reviewer 3 Report

I do not have comments for the authors after reading the new version of the manuscript.